# Integration of single photon emitters in 2D layered materials with a silicon nitride photonic chip

Frédéric Peyskens[1,2]*, Chitraleema Chakraborty[1], Muhammad Muneeb[2], Dries Van Thourhout [2] & Dirk Englund [1]

Photonic integrated circuits (PICs) enable the miniaturization of optical quantum circuits because several optic and electronic functionalities can be added on the same chip. Integrated single photon emitters (SPEs) are central building blocks for such quantum photonic circuits. SPEs embedded in 2D transition metal dichalcogenides have some unique properties that make them particularly appealing for large-scale integration. Here we report on the integration of a $WSe_2$ monolayer onto a Silicon Nitride (SiN) chip. We demonstrate the coupling of SPEs with the guided mode of a SiN waveguide and study how the on-chip single photon extraction can be maximized by interfacing the 2D-SPE with an integrated dielectric cavity. Our approach allows the use of optimized PIC platforms without the need for additional processing in the SPE host material. In combination with improved wafer-scale CVD growth of 2D materials, this approach provides a promising route towards scalable quantum photonic chips.

[1] Quantum Photonics Group, Research Laboratory of Electronics, Massachusetts Institute of Technology, Cambridge, MA 02139, USA. [2] Photonics Research Group, INTEC, Ghent University-imec, Center for Nano- and BioPhotonics, Ghent University, Technologiepark-Zwijnaarde 126, 9052 Ghent, Belgium. *email: fpeysken@mit.edu

Photonic integrated circuits (PICs) enable the miniaturizing of complex quantum optical circuits with large numbers of photonic devices connected with optimized insertion losses and phase stability[1]. Photons in a PIC are routed in a single spatial mode of a low-loss single mode waveguide, consisting of a high index core surrounded by lower index cladding materials to provide confinement of the optical mode. Spatial mode matching, which is crucial for classical and quantum interference, can be nearly perfect for such an architecture[1]. The use of PICs moreover allows integration of several functionalities on a single chip, including photonic cavities to enhance light-matter interaction, filters to block or select specific wavelengths, integrated photodetectors, etc. A central building block for such quantum photonic circuits are single photon emitters (SPEs)[2]. Over the past decade a variety of material systems have been investigated to create on-chip SPEs, including III–V quantum dots[3], carbon nanotubes[4], GaSe crystals[5], and crystal colour centers such as the NV[6] or SiV[7] centers in diamond.

Recently, SPEs were discovered in monolayer transition metal dichalcogenides (TMDCs)[8–12] and monolayer and multilayer hexagonal boron nitride (hBN)[13,14]. It has been shown that nanoscale strain engineering can be used to scale up the creation of such 2D-SPEs[15–20], but integration with a PIC has not been achieved so far. This would however alleviate some important issues met with other approaches for quantum photonic applications. First of all, techniques to transfer 2D materials or stack them by Van der Waals epitaxy to create complex heterostructures are by now getting well established, enabling easy interfacing with high quality PICs[21–23]. Secondly, it is possible to achieve very high light extraction efficiencies because the emitters are embedded in a monolayer, avoiding total internal reflection. This is a major issue for diamond and III–V based quantum technologies, where a separate photonic structure is typically made in the host material to allow efficient single photon transfer between the host and underlying PIC. This adds serious challenges because separate PICs have to be fabricated in the host material and moreover may require precise pick-and-place techniques to integrate both PICs together[6,24]. Furthermore, 2D materials can easily be integrated with electrical contacts[25] to ultimately enable all-electrical single photon generation over a broad spectrum[26] or to tune the single photon wavelength and symmetry by the quantum-confined Stark effect[27,28]. Finally, 2D materials grown with high wafer-scale uniformity are becoming widely available[29–31], such that they can be matched at the wafer level with underlying photonic circuitry. Since 2D-SPEs mainly emit in the visible, the standard silicon-on-insulator PIC platform cannot be used because it's not transparent for these wavelengths. Silicon nitride (SiN) PICs on the other hand are a useful platform for routing photons that carry quantum information since they provide low-loss transmission in the visible and are also available in a CMOS-fab[32].

Here we study the integration of a WSe₂ monolayer onto a SiN chip and demonstrate the coupling of 2D-based single photon sources with the guided mode of a SiN waveguide. We discuss how integrated cavity-emitter systems, evanescently coupled to a waveguide, should be designed to optimize single photon extraction into the waveguide. As such the full potential of a high quality and CMOS-compatible PIC platform can be exploited without the need for stringent processing in the host material itself. In combination with wafer-scale growth of 2D materials, this provides a promising route towards scaling of quantum photonic circuits.

## Results

### Device overview. 
Figure 1a shows a schematic of the device. A mechanically exfoliated WSe₂ flake is transferred by dry transfer onto a single mode SiN waveguide. After transfer, the sample was placed in an optical cryostat from Montana instruments and cooled down to 3.9 K. Photoluminescence (PL) from the WSe₂ can either couple to free-space radiation or to the guided mode of the waveguide. The radiation to free-space is collected by a top objective with NA = 0.65, while the waveguide-coupled PL is captured by a lensed fiber, aligned to the output facet of the waveguide. An impression of the fiber-coupled chip and a microscope image of the integrated WSe₂ flake are depicted in Fig. 1b. The typical 1/e single photon propagation length for our devices is 0.5–1 cm (≈4–10 dB cm⁻¹). See Supplementary Note 1–3 for more information on the device fabrication and experimental setup, as well as a plot of a typical spectrum of the flake showing the neutral exciton peak around 710 nm with the broad delocalized neutral exciton defect band.

To maximize the count rate of an integrated single photon source, the fraction $\eta_{wg}$ of total PL that couples to the waveguide mode should be as close as possible to one. It is, however, impossible to achieve this with the simple waveguide geometry shown in Fig. 1a, but interaction with a cavity can significantly boost the overall coupling rate to the guided mode. As an extension of our experimental results we will therefore investigate for which cavity parameters near-unity waveguide extraction efficiencies can be obtained. An essential parameter in this calculation is the cavity-emitter coupling, which critically depends on the dipole moment strength of the integrated 2D-based emitter. For realistic estimates of this value, we will assess it from our experiments. As such we can get a clear overview of which cavity Q–factors and mode volumes $V_c$ are required to maximize single photon extraction.

### On-chip quantum emitters. 
Figure 2 summarizes PL measurements on the flake. The excitation beam ($\lambda = 532$ nm) can be scanned over the sample through the top window of the cryostat by a set of two galvo-mirrors. The regions that light up in the PL scan of Fig. 2b, match with the area covered by the flake in the scanning confocal image of Fig. 2a. We will investigate five different spots on the flake, labeled S1 through S5. The spectra for two positions off the waveguide (S1 and S2) are shown in Fig. 2e. Spot S1 exhibits only two prominent peaks, which are relatively weaker compared to the spot S2 peaks. Spot S2 contains several narrower peaks with FWHM on the order of 3 meV in the 1.65–1.7 eV spectral region. This result is similar to observations made by Tonndorf et al.[8]. Spot S2 appears near a spatial non-uniformity in the flake (Fig. 2a), which could be due to e.g. a wrinkle in the monolayer, a crack in the material or a transition between a mono- and bi-layer. Such spatial non-uniformities usually lead to strong strain-gradient regions. Previous reports have shown that such regions are usually associated with the appearance of localized bright spots containing narrow linewidth emitters in TMDC monolayers[15–17]. As such, the most likely mechanism for the appearance of multiple narrower peaks in the spectrum of spot S2 is strain (see also furtheron for the spectra of spots near the waveguide ridge). For all spectra in Fig. 2, the excitation power was set to 25 nW with an integration time of 60 s. Because the excitation power was low, the FWHM was not affected by power broadening. Spectral wandering during the long integration time could, however, result in inhomogeneous broadening of the FWHM of the emitters, as observed in earlier studies[12].

The areas near spots S3, S4, and S5 exhibit brighter fluorescence compared to the surrounding region (see Fig. 2b) and are all located in the vicinity of the waveguide (region between the white dotted lines). This is similar to recent reports in which bright emission of a TMDC monolayer was observed at

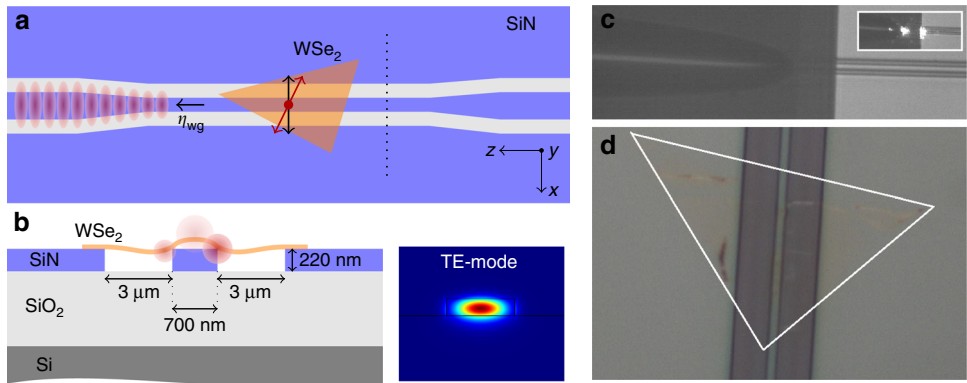

**Fig. 1** Integrated WSe$_2$ quantum emitters. **a** Top view of the device: a WSe$_2$ flake is integrated on a 220 nm thick single mode SiN waveguide, separated by 2 air trenches from the bulk SiN. The waveguide ends are tapered to allow easier coupling with a lensed fiber. The orientation of the dipole moment of the WSe$_2$ emitters (red arrow) is random with respect to the quasi-TE polarization (approximately aligned along $x-$direction) of the fundamental waveguide mode (black arrow). A fraction $\eta_{wg}$ of the total emission couples into the left-propagating waveguide mode (represented by red shaded areas). **b** Cross-section of the sample. The width of the air trenches and waveguide is 3 μm and 700 nm respectively. The generated PL of emitters near the waveguide couples both to free-space and to the waveguide (red shaded circles). A cross-sectional mode profile (at $\lambda = 750$ nm) of the waveguide, taken along the dotted black line in the top figure, is shown as well. **c** Impression of the fiber-coupled chip (inset shows light coupling from the fiber to the chip). The tapered lensed fiber is a standard SM630 fiber from Thorlabs with a focal spot size of 2 μm and an 8 μm working distance. **d** Microscope image of SiN chip with WSe$_2$ transferred on waveguide region. The flake is highlighted by the white triangle

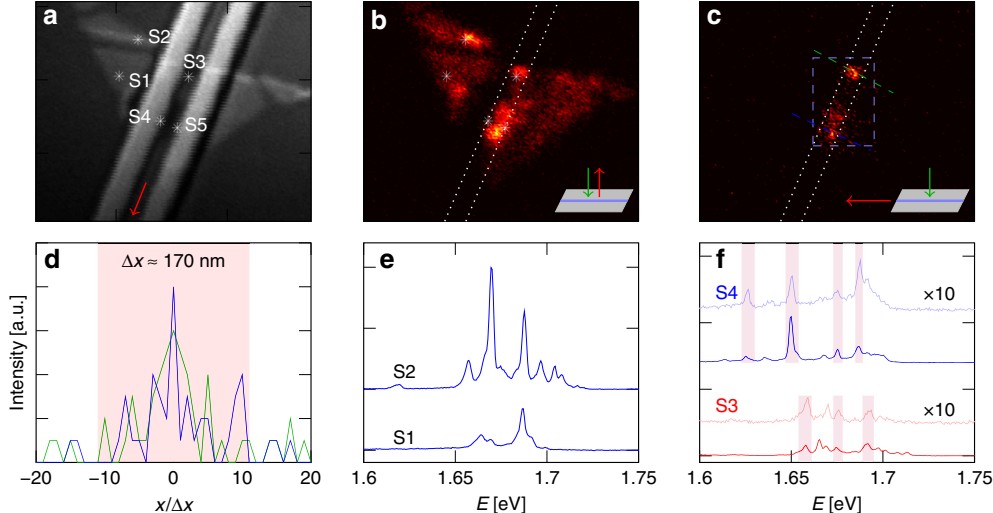

**Fig. 2** Waveguide-coupled WSe$_2$ quantum emitters. **a** Confocal laser scan ($\lambda = 532$ nm) of the relevant sample area. Spots S1 and S2 are spots off the waveguide, while spots S3 to S5 mark positions on the waveguide. The red arrow indicates the direction along which the fiber-coupled light is collected. **b** Confocal PL scan, by scanning the excitation beam over the sample from the top and collecting the PL from the top. **c** Waveguide PL scan, by scanning the excitation beam over the sample from the top and collecting the PL through the fiber. The white dotted lines mark the waveguide position. **d** Line scan along the the green and blue lines in Figure **c**, highlighting the estimated spatial region coupled to the waveguide (shaded red region). **e** PL spectra from spots S1 and S2, collected from the top. **f** PL spectra from spots S3 (red) and S4 (blue), collected from the top (solid color) and through the fiber (shaded color). Matching peaks are highlighted by shaded purple regions. Where necessary, the spectrum baseline is shifted for improved visualization. The waveguide-coupled spectra are multiplied by 10. The excitation power for all PL spectra was $P_e \approx 25$ nW

positions where the material was bend over a nanopillar and hints to the presence of strain-induced emitters coupled to the waveguide[16,17]. To confirm that these spots are indeed coupled to the waveguide, we scan the excitation beam from the top, but collect the PL through the lensed fiber and indeed observe that only the waveguide region lights up (Fig. 2c). A small offset in the piezo position of the fiber from the waveguide results in an immediate loss of the signal, further confirming that we indeed collect light originating from the waveguide. The integrated intensity near the center of the waveguide is in general higher, which could be attributed to the fact that the electromagnetic

overlap with the waveguide mode is higher near the center. As such, more radiation from the 2D material can couple into the waveguide. When the emitters are not located on the waveguide, it is interesting to estimate how far they can be away from the waveguide core and still generate PL that can couple into the waveguide. Figure 2d shows a line scan along two lines perpendicular to the waveguide to estimate the spatial extent over which the PL can still be coupled. Emitters located up to 1.9 μm on either side of the waveguide can couple into the waveguide. A closer examination of the confocal and waveguide-coupled spectra of spots S3 and S4 is shown in Fig. 2f. The spectra

feature several narrow lines, with a typical linewidth ranging between 2.5 and 4 meV. This linewidth can be significantly broadened by the immediate surrounding of the WSe$_2$ (e.g. surface charges in the SiO$_2$ and SiN), but the broadening can be partially alleviated by encapsulation with hBN[33,34]. A comparison between the spectrum of spot S1 and the other spots moreover shows more peaks near the waveguide or cracks in the sample, substantiating the argument that the emitters are indeed strain-induced. Data from a hyperspectral scan of the blue-dashed area in Fig. 2c, containing info on the spectral distribution of the PL and an estimation on the number of peaks, are included in Supplementary Note 4.

A comparison of the confocal and waveguide-coupled spectra shows that not all peaks appearing in the confocal spectra are present in the waveguide-coupled spectra. This can be understood from the fact that the coupling between the waveguide mode $\mathbf{E}_{wg}$ (quasi-TE-mode in our case) and the dipole moment of the quantum emitter $\mathbf{p}_d$ scales according to $\mathbf{E}_{wg} \cdot \mathbf{p}_d \propto \cos\theta_d$, with $\theta_d$ the angle between $\mathbf{E}_{wg}$ (black arrow in Fig. 1a) and $\mathbf{p}_d$ (red arrow in Fig. 1a). Hence, when $\theta_d \to \frac{\pi}{2}$, the coupling vanishes. According to numerical simulations with Lumerical FDTD solutions, about $\eta_{wg} = 7.3\%$ of the total power radiated by a dipole (at $E = 1.63$ eV) with $\theta_d = 0$ and centered on the top surface of the waveguide couples in the left-propagating guided TE-mode. For the same dipole emitter, $\eta_{NA} \approx 6.5\%$ radiates upwards in an NA = 0.65. A dipole at the same position on the waveguide but with $\theta_d = \frac{\pi}{2}$ does not radiate into the TE-mode (as expected by the $\cos\theta_d$ behaviour), while emitting $\approx 7.3\%$ upwards in an NA = 0.65. So regardless of the orientation of the dipole, we expect about 7% of the total radiation to be captured in an NA of 0.65, while the light captured by the waveguide heavily depends on $\theta_d$. As such, the large spread in relative strength between the confocal and waveguide-coupled signal of a certain peak stems from the fact that their ratio scales as $\eta_{wg}/\eta_{NA} \propto \cos\theta_d$. The relative strength between different peaks depends both on the dipole polarization as well as on the absolute dipole moment of the emitter.

**Waveguide-coupled single photon source**. We will now focus on spot S5 of Fig. 2a and investigate the quantum nature of the observed emitters in more detail. The confocal and waveguide-coupled spectrum of spot S5 are shown in Fig. 3a. We observe a few peaks recurring in both the confocal and waveguide spectrum, confirming that these emitters are indeed coupled to the waveguide. A prominent and isolated waveguide-coupled peak (FWHM ≈3 meV) appears around 1.64 eV (756.5 nm). It has been shown that the PL of 2D-based quantum dots can be enhanced when the excitation laser wavelength is tuned close to the free excitonic resonance[9]. When we scan the excitation wavelength with a tunable Ti:saph laser around the free exciton wavelength, we also find a considerable increase in peak count rate and reduction in background compared to excitation with $\lambda$ = 532 nm for the same excitation power (see inset Fig. 3a). An excitation wavelength of $\lambda = 702$ nm provided the most optimal ratio between peak count rate and background, and hence the emitter was pumped at this wavelength for all subsequent experiments.

A 750 nm longpass filter (gray shaded area in Fig. 3a) was used to spectrally isolate the 1.64 eV peak from the broad PL emission around 1.7 eV before the beam hits the Single Photon Detectors (SPDs). As such, the major contribution to the SPD count stems from the 1.64 eV peak and we can perform a $g^{(2)}$−measurement to investigate whether single photons are emitted by this emitter. Due to the lower count rates of the waveguide-coupled PL, we use the free-space collected PL for the $g^{(2)}$−measurement. Based on

the spectrum we assess that the peak of interest (at 1.64 eV) contributes a fraction of about $\rho = 0.76$ to the total signal while the rest is due to uncorrelated background. The raw normalized coincidence counts without any background correction are reported in the Supplementary Note 5, while the plot in Fig. 3b shows the background-corrected $g^{(2)}(\tau)$−curve, on which moreover a running average is applied to reduce the noise on the data. The background corrected $g_{BC}^{(2)}(\tau)$ value can be calculated according to $g_{BC}^{(2)}(\tau) = (g^{(2)}(\tau) - (1 - \rho^2))/\rho^2$[35]. See Supplementary Note 5 for more details on the background correction and running average. Fitting the background-corrected data to the equation $g_f^{(2)}(\tau) = 1 - A\exp(-|\tau|/\tau_f)$ yields $g_f^{(2)}(0) = 1 - A = 0.47$ and $\tau_f = 7.99$ ns[36]. The minimum value in the background-corrected data without averaging is about 0.03, which would hint to almost perfect single photon emission. The fitted rise time $\tau_f =$ 7.99 ns is a lower limit for the PL decay time and is in the same order of magnitude as previously reported values for WSe$_2$[8]. The clear anti-bunching dip with a background corrected $g^{(2)}(0) < 0.5$ confirms that the emitter indeed emits single photons.

A generic two-level system moreover exhibits saturation of the PL emission when the excitation rate increases, and this has been observed for WSe$_2$ emitters before[8–10,12]. The PL saturation for our waveguide-coupled quantum emitter is shown in Fig. 3b. A fit of the PL intensity $I = I_s(P_e/(P_e + P_s))$ as a function of excitation power $P_e$ yields a saturation power of $P_s \approx 500$ nW (at $\lambda = 702$ nm) and a saturation intensity of $I_s \approx 100$ kHz. The excitation efficiency of the emitter will, however, depend on the orientation between the dipole moment of the quantum emitter $\beta_d$ and the excitation polarization $\beta_e$ and will hence affect the measured intensity. We therefore perform polarization-dependent transmission measurements to determine $\Delta\beta = \beta_d - \beta_e$. The normalized transmitted emitter count rate to SPD1 as a function of the polarization-rotating half-wave plate angle $\alpha$ is shown in Fig. 3d. By fitting this count rate one can determine $\Delta\beta$ and eventually assess the saturation count rate of the single photon source. When corrected for transmission and collection efficiencies of the system, the total saturation intensity is about 3 MHz (to all modes, guided and non-guided) while the estimated maximum waveguide-coupled count rate is about 100 kHz (see Supplementary Note 6). Further improvements consist of changes in the waveguide design[37] or interaction with plasmonic or dielectric cavities[38,39] to maximize the coupling efficiency into the guided mode and enhance non-classical light generation.

**Optimized single photon extraction and indistinguishability**. Apart from high single-photon extraction efficiency, various applications (linear optical quantum computing, quantum tele-portation, quantum networks, etc.) require the single photons to be indistinguishable (i.e. identical spatial and spectral modes)[40]. For an ideal single photon source, the product of extraction efficiency $\eta$ and indistinguishability $V$ should be $\eta V = 1$. In this section we will assess how $\eta$ and $\eta V$ of an integrated 2D quantum emitter can be optimized by cavity coupling. Figure 4a shows a schematic of the investigated platform. The emitter is coupled to a cavity with coupling strength $\Omega$, while the cavity is evanescently coupled to the waveguide with a coupling strength $\kappa$. The intrinsic decay rate of the cavity $\gamma_c$ contains both absorption losses and radiation to non-guided modes. The overall cavity decay rate (containing both intrinsic losses as well as coupling to the nearby waveguide) is given by $\gamma_p = \gamma_c + \kappa$. The rate $\gamma_e$ incorporates decay of the emitter to all modes (radiative and non-radiative) other than the cavity and $\gamma^*$ is the emitter dephasing (which describes a decay of the atomic polarization $S_x + iS_y$, without changing the decay of $S_z$ and is modeled by a coupling between $S_z$ and a high temperature heat bath; $S_{x,y,z}$ are the the

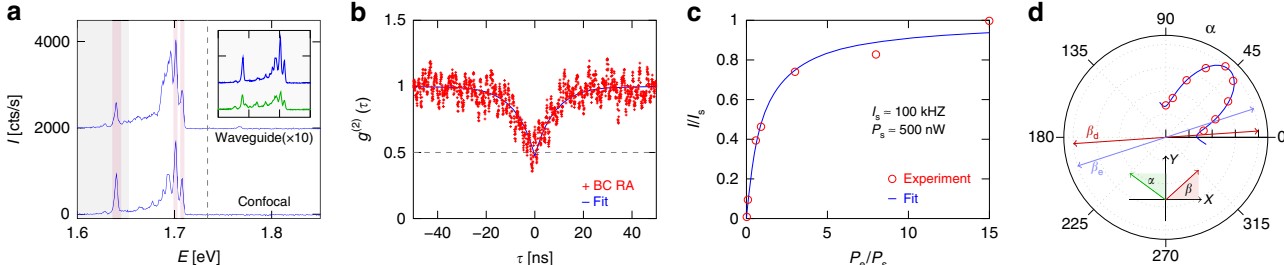

**Fig. 3** On-chip single photon emission. **a** Confocal and waveguide-coupled spectrum of spot S5, excited with $\lambda = 702$ nm. The waveguide spectrum is multiplied by 10 and offset by 2000 cts/sec for improved visualization. Common peaks are highlighted by the shaded purple regions. A 715 nm (1.73 eV) longpass filter, marked by the dashed line, was used to filter the pump. For the $g^{(2)}(\tau)$ measurement a 750 nm (1.65 eV) longpass filter (gray shaded area) was used to isolate the single emitter at 756.6 nm (1.64 eV). The inset figure shows confocal spectra obtained by either green ($\lambda = 532$ nm) excitation (green curve) or excitation with $\lambda = 702$ nm (blue curve). **b**–**d** Characterization of the 1.64 eV emitter. **b** Normalized background-corrected (BC) running average (RA) coincidence counts (red) and $g^{(2)}(\tau)$ fit (blue). **c** Measured intensity saturation (red) and fit to saturation curve (blue). **d** Normalized SPD count of the emitter (red) as a function of half-wave plate rotation angle $\alpha$ and fit to intensity transmission curve (blue); $\alpha = 0$ corresponds to a half-wave plate fast axis along the $Y$−direction. See Supplementary Fig. 1 for orientation of the half-wave plate with respect to the $(X, Y, Z)$ axes. Based on the fit, the difference in polarization angle between the PL ($\beta_d$) and excitation ($\beta_e$) beam can be extracted; $\beta = 0$ corresponds to a polarization along the $X$−axis

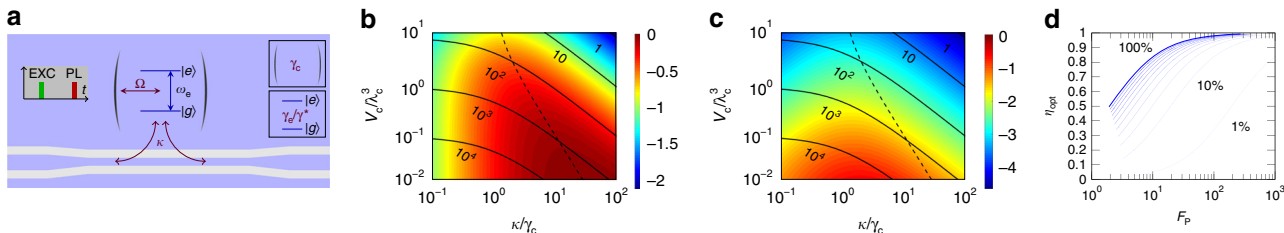

**Fig. 4** Integrated cavity-emitter system. **a** Schematic of an integrated cavity-emitter system, evanescently coupled to a single mode waveguide. The coupling rate between an emitter with frequency $\omega_e$ and a cavity with resonance frequency $\omega_c$ is given by $\Omega$. The decay rate from the cavity-emitter system to the guided mode is $\kappa$, while the other decay channels of the emitter and cavity are given by $\gamma_e$ and $\gamma_c$, respectively. The emitter dephasing is described by $\gamma^\star$. The system is excited (EXC) by a short pulse and subsequently the single photon PL is collected. **b**, **c** Single photon extraction efficiency $\eta$ (**b**) and extraction-indistinguishability product $\eta V$ (**c**) as a function of cavity mode volume $V_c$ and cavity decay rate $\kappa$. The black solid lines represent lines of constant Purcell factor $F_P$, while the black dashed line represents $(V_c, \kappa)$ combinations for which $\eta$ is maximal. The parameter values used to generate plots **b**, **c** are $Q_i = 10000$, $\Gamma = 3$ MHz, $\gamma_e = 300$ MHz, $\cos^2\theta_d = 1/2$ (i.e. average over different orientations of the quantum emitter), $\gamma^\star = 100$ GHz[44], $n_d = 4$ and $\lambda_c = \lambda_0/n_{eff}$ with a free-space wavelength of $\lambda_0 = 750$ nm and an effective refractive index of $n_{eff} = 1.6$ for the fundamental TE-mode (which is calculated using a commercial FDTD solver from Lumerical). Both plots are on a $\log_{10}$ color scale, i.e. 0 corresponds to perfect $\eta = 1$ or $\eta V = 1$. **d** Optimum $\eta_{opt}$ (evaluated at $(V_c, \kappa)$ combinations for which $\eta$ is maximal) as a function of $F_P$ for different values of $\Gamma$ ranging from $0.01\gamma_e$ to $\gamma_e$ (different quantum yields)

Pauli matrices[41]). For our calculations we assume the emitter is resonant with the cavity ($\omega_e = \omega_c$) and is initialized in the excited state by a short excitation pulse (EXC) with no photons present in the cavity. The master equation governing the dynamics of this system is discussed in Supplementary Note 7. In the regime where $\gamma^\star \ll \gamma_e + \gamma_p$ (which should be satisfied for low temperatures and moderate $Q$–factor cavities), the single photon extraction efficiency into the guided mode ($\eta$) is given by

$$\eta = \frac{\kappa}{(\gamma_e + \gamma_c + \kappa)\left(1 + \frac{\gamma_e(\gamma_c + \kappa)}{4\Omega^2}\right)}. \tag{1}$$

The expressions for the indistinguishability $V$ of photons coupled into the guided mode, as derived by Grange et al.[40], depend on the regime within which the system falls (see Supplementary Note 7). To assess $\eta$ and $\eta V$ (as shown in Fig. 4b, c), we first need to determine the different coupling strengths. The coupling constant $\Omega$ depends on the cavity mode volume $V_c$ through $\Omega^2 = \frac{3\pi c^3}{2n_d\omega_c^2}\cos^2\theta_d\left(\frac{\Gamma}{V_c}\right)$, with $\Gamma$ the free-space radiative decay rate in a uniform dielectric with index $n_d$, and $\theta_d$ the angle between the emitter dipole moment and the cavity field. For our

calculations we assume $n_d$ is the refractive index of a WSe$_2$ monolayer ($n_d = 4$)[42]. In our case, the radiative decay rate to non-guided modes will usually differ from $\Gamma$ due to the non-uniform dielectric environment and may furthermore be influenced by the vicinity of the dielectric cavity, but as a simplifying assumption we set $\Gamma \approx \gamma_r$ with $\gamma_r$ the radiative decay rate determined from our experiment, i.e. $\gamma_r \approx 3$ MHz. Numerical simulations of dipole emission near a waveguide show that the total radiated dipole power (with polarization parallel to the top surface of the waveguide) is on the same order of magnitude as what the dipole would radiate in a homogeneous dielectric, so in a first approximation this is a valid assumption. To take into account different polarizations of the quantum emitter, we assume an average value for $\cos^2\theta_d$ over all possible orientations $\theta_d$, i.e. $< \cos^2\theta_d > = 1/2$. The decay rate $\gamma_e$ also contains contributions to non-radiative modes ($\gamma_e = \gamma_r + \gamma_{nr}$), and can be approximated by $\gamma_e = \gamma_r/\xi$ with $\xi$ the quantum yield of the monolayer. Strain-induced quantum emitters in WSe$_2$ are reported to have a typical quantum yield of 1%[43], so we take $\gamma_e \approx 300$ MHz for our calculations. It is important to note that the quantum yield of these emitters can however vary significantly depending on growth conditions. As such the 1% is only a first approximation. The effect of different quantum yields will be described furtheron.

The final parameter is $\kappa$, which we express through the intrinsic cavity quality factor $Q_i$ as $\kappa = \chi\gamma_c = \chi\left(\frac{\omega_c}{2Q_i}\right) = \frac{\omega_c}{2Q_\kappa}$ such that the loaded quality factor of the cavity is given by $Q = \left(Q_i^{-1} + Q_\kappa^{-1}\right)^{-1} = Q_i/(1+\chi)$. We use $Q_i = 10000$ for our calculations. The above parameter values are now used to estimate how $\eta$ and $\eta V$ can be improved through cavity-assisted interaction as a function of the normalized cavity mode volume $V_c/\lambda_c^3$ and waveguide-cavity coupling $\chi = \kappa/\gamma_c$ (Fig. 4b, c). The solid black lines represent lines of constant Purcell factor $F_P = \frac{3}{4\pi^2} Q\left(\frac{\lambda_c^3}{V_c}\right)$, while the dashed black line represents the $(V_c, \kappa)$ combinations for which $\eta$ is optimized. For a given mode volume $V_c$ (i.e. $\Omega$), the coupling rate $\kappa$ that maximizes $\eta$ is given by

$$\kappa_{opt} = \gamma_c \sqrt{\left(1 + \frac{\gamma_e}{\gamma_c}\right)\left(1 + \frac{4\Omega^2}{\gamma_e\gamma_c}\right)}. \qquad (2)$$

For this value of $\kappa$, the optimum $\eta \approx \mathcal{C}/\left(1 + \sqrt{1+\mathcal{C}}\right)^2$ if $\gamma_e < \gamma_c$, with $\mathcal{C} = 4\Omega^2/(\gamma_e\gamma_c) \propto \xi Q_i(\lambda_c^3/V_c)$. As such, near-unity extraction requires a high intrinsic quality factor (while the loaded $Q$ can be much lower), high quantum efficiency and small mode volume. The intersection of the $F_P = 100$ line with $\kappa_{opt}$ yields $\eta \approx 34\%$ for $\kappa = 2\gamma_c$ ($Q = 3333$) and $V_c = 2.45\lambda_c^3$. For these parameter values, $\eta V$ is only 0.2% however. To achieve high $\eta V$ one typically needs much smaller $V_c$ because the cooperativity $\mathcal{C}$ has to overcome the emitter dephasing $\gamma^{*44}$. If we decrease $V_c$ to $V_c = 0.01\lambda_c^3$, then a maximum $\eta V \approx 25\%$ is achieved for $\kappa = 2.05\gamma_c$ ($Q \approx 3280$). A near-unity extraction ($\eta = 93\%$) can be achieved for $\kappa = 29\gamma_c$ ($Q \approx 333$) and $V_c = 0.01\lambda_c^3$ ($F_P \approx 2530$), with $\eta V \approx 6.8\%$. By using the ultrasmall mode volume nanocavities reported in[45], we could hence achieve near perfect single photon extraction, even for a very low quantum yield emitter. However, the required cavity Purcell factor is still large. In order to achieve higher $\eta$ for smaller $F_P$ one can aim to increase the quantum yield as shown in Fig. 4d, which depicts $\eta_{opt}$ (i.e. $\eta$ evaluated at $(V_c, \kappa)$ combinations for which $\eta$ is maximal) as a function of $F_P$ and $\xi$. For near-unity quantum yield, $\eta$ already reaches 84% for a moderate Purcell factor of $F_P = 10$, while $\eta = 98\%$ for $F_P = 100$. Nevertheless, the corresponding $\eta V$ product is still far from the desired unity value. In most realistic cases, the system will be in the bad cavity limit ($\gamma_P > \gamma_e + \gamma^*$), and achieving high $\eta V$ will require the Purcell factor to satisfy $F_P \gg (1/\xi + \gamma^*/\Gamma)$ (see Supplementary Note 7 for formulas of $V$ in different limits). To relax the constraints on $F_P$ one should hence aim to reduce the ratio $\gamma^*/\Gamma$ or increase $\xi$. However, usually $\gamma^*/\Gamma > 1/\xi$ so increasing $\xi$ will have little effect as long as the dephasing rate is high. This analysis can be repeated for any dielectric cavity-emitter system that is evanescently coupled to the waveguide and as such can guide future design efforts to optimize single photon extraction and indistinguishability of photons coupled into the guided mode of the waveguide.

## Discussion

In conclusion we have demonstrated that integration of a WSe$_2$ monolayer onto a SiN waveguide results in quantum emitters evanescently coupled to the waveguide. Second-order correlation measurements on a spectrally isolated quantum emitter confirm that single photons are emitted with a waveguide-coupled saturation count rate of 100 kHz. These results confirm previous claims that strain-induced quantum emitters could be coupled to photonic structures[16,17]. A numerical analysis on the optimization of single photon extraction and indistinguishability using integrated dielectric cavity-emitter systems indicates that near-unity single photon extraction can be achieved, even for low quantum yield emitters. The presented approach for integration of strain-induced TMDC-based SPEs retains the favorable attributes of SiN PICs without the need for stringent processing in the quantum emitter host material itself. Recent progress in wafer-scale growth and patterning of identical 2D-material based devices[29–31] provides a promising route in combination with our waveguide-coupled 2D-SPEs to scale up quantum photonic circuits.

## Data availability
The data that support the findings of this study are available from the corresponding author upon reasonable request.

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

## Acknowledgements

We acknowledge Liesbet Van Landschoot and Steven Verstuyft for processing of the SiN chips, Hyowon Moon for building the confocal setup, and Noel Wan for help in making the custom vacuum fiber feedthrough and installing the fiber-coupling unit. F.P. acknowledges support from an FWO (Fonds voor Wetenschappelijk Onderzoek—Vlaanderen) postdoctoral fellowship. F.P. and C.C. acknowledge partial support from the Army Research Office MURI (Ab-Initio Solid-State Quantum Materials) grant number W911NF-18-1- 0431 and D.E. from the Army Research Laboratory Center for Distributed Quantum Information (CDQI).

## Author contributions

F.P. designed the chip, conducted the experiments, analyzed the data and developed the theoretical model. C.C. performed the transfer of the 2D material and helped with the experiments. M.M. fabricated the sample. D.V.T. and D.E. supervised the work.

## Competing interests

The authors declare no competing interests.
