## [Peer Review File · Nature Communications]

Reviewers' comments:

Reviewer #1 (Remarks to the Author):

The authors have shown the coupling of SPEs with the guided mode of a SiN waveguide and studied how the on-chip single photon maximized by interfacing with an integrated dielectric cavity. Authors have tried to integrate SPE from WSe₂ with SiN chips as Photonic integrated circuits (PICs) and claim that such integration has never been done. One of the advantages for TMDC based SPEs involve single photon efficiency without worrying about total internal reflection as the SPEs are embedded in the monolayer itself, on the contrary, other diamonds based or III-V based emitters need a host for efficient transfer to photonic integrated circuits. Further 2d materials can be integrated with the electrical contacts as well. This work is interesting but quality of presentation can be improved further. I recommend a major revision. I have the following comments and suggestions for improvements.

1. In figure 1, How the lensed fiber is attached to the waveguide? What are the characteristics of the attached fiber with the specification is missing?
2. How much are cavity quality factor and mode volume? How does it depend on the SiN width? How does it compare to those of Metal insulator metal (MIM)? What about the propagation length of the single photons?
3. In Figure 2 (d), What's the significance of the line scan? It is not discussed?
4. In Figure 4 c color scales are not visible. Please explain about the emitter dephasing. How is the N_{eff} for the TE mode calculated?
5. Authors have claimed Indistinguishability of the photon using analytical model, however, nowadays it is routinely measured using Hong–Ou–Mandel (HOM) set up. I suggest the authors verify their analytical results from HOM measurements.

6. For the sake of comparison, extraction and coupling efficiency of plasmonic waveguides, such as metallic wires, metallic nanogaps, and metal insulator metal waveguides should also be calculated.

7. The authors claim that they are the first to show the integration of SPS with photonic circuits. How are the results different from the following papers.

a. Deterministic Integration of Single Photon Sources in Silicon-Based Photonic Circuits

Iman Esmail Zadeh*†Ali W. Elshaari*†‡Klaus D. Jöns†‡Andreas Fognini†Dan Dalacu§Philip J. Poole§Michael E. Reimer||Val Zwiller, *Nano Lett.* 2016 16 42289-2294, 2016.
<https://doi.org/10.1021/acs.nanolett.5b04709>.

b. Strain-Tunable Quantum Integrated Photonics, Ali W. Elshaari*Efe BüyüközerIman Esmail ZadehThomas LettnerPeng ZhaoEva SchöllSamuel GygerMichael E. ReimerDan DalacuPhilip J. PooleKlaus D. JönsVal Zwiller, *Nano Lett.* 2018 18 127969-7976, 2018.
<https://doi.org/10.1021/acs.nanolett.8b03937>.

c. Near-Unity Indistinguishability Single Photon Source for Large-Scale Integrated Quantum Optics, Łukasz Dusanowski, Soon-Hong Kwon, Christian Schneider, and Sven Höfling, *Phys. Rev. Lett.* 122, 173602.

Furthermore, the authors should also include at the place "It has been shown that nanoscale strain engineering can be used to scale up the creation of such 2D-SPEs [15, 16, 17]....., some of the relevant recent papers on strain-induced single-photon generation from WSe₂ monolayers and the deterministic coupling of single photons from WSe₂ with plasmonic cavities as follows.

a. Kern, J.; Niehues, I.; Tonndorf, P.; Schmidt, R.; Wigger, D.; Schneider, R.; Stiehm, T.; Michaelis de Vasconcellos, S.; Reiter, D. E.; Kuhn, T.; Bratschitsch, R.; et al. Nanoscale Positioning of Single-Photon Emitters in Atomically Thin WSe₂. *Adv. Mater.* 2016, 28,

7101–7105.

b. Laxmi Narayan Tripathi, Oliver Iff, Simon Betzold, Łukasz Dusanowski, Monika Emmerling, Kihwan Moon, Young Jin Lee, Soon-Hong Kwon, Sven Höfling, and Christian Schneider *ACS Photonics* 2018 5 (5), 1919-1926. DOI: 10.1021/acsp Photonics.7b01053.

c. Oliver Iff, Nils Lundt, Simon Betzold, Laxmi Narayan Tripathi, Monika Emmerling, Sefaattin Tongay, Young Jin Lee, Soon-Hong Kwon, Sven Höfling, and Christian Schneider, "Deterministic

coupling of quantum emitters in WSe2 monolayers to plasmonic nanocavities," Opt. Express 26, 25944-25951 (2018).

Reviewer #2 (Remarks to the Author):

In the paper, the authors reported coupling of single quantum emitters in WSe2 to SiN waveguide. They also analyzed performance of a cavity integrated emitter system. I do not find any novelty in the research. WSe2 emitters are very well-studied and integrating with SiN WG is extremely simple. Integration with cavities (experimentally) would have been a significant milestone (due to spectral matching needed), but only theoretical analysis is provided, which is similar to any other emitters. In fact, the authors have published research on similar analysis in the past. Hence, I do not recommend the paper for publication.

Reviewer #3 (Remarks to the Author):

In this article, the authors discuss the coupling of single photon emitters originating from a monolayer of WSe2 to a SiN waveguide, a primary component in photonic integrated circuits. Despite a potentially low quantum yield from the single photon emitters, the authors calculate the potential for a high extraction efficiency while indistinguishability remains limited by the SPE quality. This manuscript highlights the ability for monolayer TMDs to be integrated into well-developed PIC processes with a simple transfer step and gives direction for further improvement. Recently there has been a lot of interests in single photon emitters in TMDs and this article is timely for that.

1. The authors show the spectra for spots which specifically contain quantum emitters, but give no spectra off the spots to compare with, could this be included in the supplementary?

2. For spot S2, the authors claim that this is near the monolayer edge, however upon examining the optical image closely, one can clearly see that there is a fold in the WSe2 monolayer and it looks more like a bilayer (perhaps with a twist angle) and an area of high strain which likely is the main contribution to this peak. The authors later comment on this on page 4 without making an overall

reference or at small explanation as to why strain contributes, while misleading the reader earlier in the text. Could the authors at least expand on this and give good references, considering this is likely the main mechanism of the quantum emitters that they are coupling to?

3. Additionally, with the white dots covering the exact area of the spots examined, it is hard to tell from Figure 2b what the integrated intensity at the exact spots looks like compared to nearby or the background. Finally, from Figure 2, it seems to be that the highest intensity in the waveguide PL scan (c) comes from areas not over the spots (sans S3). Could the authors comment on this?

4. In the main text γ_{p} is completely left undefined, and this happens to a few other variables as well. Although many of these variables are defined in the supplementary, there should be at least some clarification in the main text, given that there are a particularly large number of variables in this text and it is incredibly easy for the reader to become confused, especially those from a broader audience as expected from a journal like Nature Communications.

5. The authors use spot 5 to examine the quality of their emitter/waveguide, why specifically this spot?

6. The authors used a reported value for the quantum yield in monolayer WSe₂ to further calculate a decay rate of the emitter to all modes, this value is quoted as 3%. However, this is a vastly incorrect value to use as it takes into account the entire PL spectrum as discussed in reference 39, not the individual emitter. Secondly, in reference 39 the material is purchased from 2 separate companies which utilize CVT to produce the single crystals. These crystals can vary dramatically in quality in a singular batch (there are at least 4 main growth parameters which govern this process) making any comparison of quality to this work impossible as QY's could vary by orders of magnitude between samples extracted from different crystals in the same batch. This is not at all hinted in the calculation or the Supplementary.

First of all we would like to thank the Reviewers for their feedback on the manuscript. Reviewer 1 states our work is interesting, but recommends a major revision to mainly improve the presentation of the results. Reviewer 2 claims the results are not novel enough and does not recommend publication. Reviewer 3 states that there has recently been a lot of interest in single photon emitters in TMDs and that our article is timely given the current interest in the field. We have thoroughly revised our manuscript and believe that addressing the raised comments enabled us to substantially improve the manuscript. A detailed response to all concerns can be found below.

1 Reviewer 1

The authors have shown the coupling of SPEs with the guided mode of a SiN waveguide and studied how the on-chip single photon maximized by interfacing with an integrated dielectric cavity. Authors have tried to integrate SPE from WSe2 with SiN chips as Photonic integrated circuits (PICs) and claim that such integration has never been done. One of the advantages for TMDC based SPEs involve single photon efficiency without worrying about total internal reflection as the SPEs are embedded in the monolayer itself, on the contrary, other diamonds based or III-V based emitters need a host for efficient transfer to photonic integrated circuits. Further 2d materials can be integrated with the electrical contacts as well. This work is interesting but quality of presentation can be improved further. I recommend a major revision. I have the following comments and suggestions for improvements.

- Question: In figure 1, How the lensed fiber is attached to the waveguide? What are the characteristics of the attached fiber with the specification is missing?
 - Answer: The lensed fiber is brought in close proximity of the waveguide, but not attached to it. The tapered lensed fiber has a focal spot size of $2\mu\text{m}$ with a working distance of $8\mu\text{m}$, and the fiber type is a standard SM630 fiber from Thorlabs. We have added the fiber characteristics to the caption of Figure 1 in the revised manuscript.
- Question: How much are cavity quality factor and mode volume? How does it depend on the SiN width? How does it compare to those of Metal insulator metal (MIM)? What about the propagation length of the single photons?

- Answer: The modes of a normal SiN waveguide are not bounded in 3D as they would be for a genuine dielectric resonator. They are confined to the core of the waveguide in the cross-sectional plane, but are propagating along the waveguide direction. Hence, a quality factor or mode volume cannot be defined for such a waveguide. The SiN waveguide width is mainly chosen to keep the waveguide single mode for the wavelength region in which the emitters radiate, as this is the most optimal in terms of energy transport. Moreover, it is important that the spatial mode profile of each photon is the same if one wants to interfere the photons on-chip. The typical propagation loss of our e-beam fabricated waveguides ranges between 1 and 10 dB/cm (corresponding to single photon propagation lengths between 4.34 cm and 0.43 cm). For a typical MIM structure [1], the propagation length is about 10 μ m at 750 nm. So typically the single photon propagation length of our devices is about 1000 larger than that of an MIM structure. We have added a statement on the single photon propagation length in our revised manuscript.
3.
 - Question: In Figure 2 (d), What's the significance of the line scan? It is not discussed?
 - Answer: In the manuscript we briefly discuss that *Figure 2(d) shows a line scan along two lines perpendicular to the waveguide to estimate the spatial extent over which the PL can couple into the waveguide.* We just wanted to check that when the emitters are not located on the waveguide, how far away they could be from the waveguide core and still couple to the guided mode of the waveguide. We added an extra statement on this in the revised manuscript.
 4.
 - Question: In Figure 4 c color scales are not visible. Please explain about the emitter dephasing. How is the N eff for the TE mode calculated?
 - Answer: We have adapted the color scales of Figure 4 to increase visibility. The emitter dephasing in our model describes a decay of the atomic polarization $S_x + iS_y$, without changing the decay of S_z . This process is modeled by a coupling between S_z and a high temperature heat bath. [2] The effective index n_{eff} is calculated using a commercial electromagnetic mode solver (Lumerical FDTD solutions). We added these statements in the revised manuscript as well.
 5.
 - Question: Authors have claimed Indistinguishability of the photon using analytical model, however, nowadays it is routinely measured using Hong-Ou-Mandel (HOM) set up. I suggest the authors verify their analytical results from HOM measurements.
 - Answer: There are a few issues to perform the suggested HOM measurement. First of all, the count rate of our emitters is still relatively low (e.g. about 1000 cts/sec for isolated peak in spot S5), which would require a very long measurement time. Based on the fitted lifetime of 8 ns for our emitters and assuming a dephasing time on the order of 10 ps (as reported in Reference [3]), we have $T_2/2T_1 \approx 0.06\%$ so we are quite far from the Fourier transform limit. It was already very hard to achieve a good $g^2(0)$ measurement due to the low count rate. A significant indistinguishability measurement would furthermore require substantial spectral filtering to isolate the emitter, which would even more reduce the count rate. The discussion of indistinguishability is mainly linked to the use of cavities, which could considerably boost the count rates and improve on the overall indistinguishability. While the single photon nature of our emitters is confirmed through second order correlation measurements, we reckon a relevant HOM measurement would be technically very hard and moreover not yield a meaningful outcome because of the really low estimated $T_2/2T_1$ and the low count rate.
 6.
 - Question: For the sake of comparison, extraction and coupling efficiency of plasmonic

waveguides, such as metallic wires, metallic nanogaps, and metal insulator metal waveguides should also be calculated.

- Answer: We added a new section to the Supplementary Information (Comparison with metallic nanostructures) with additional information on coupling efficiencies between quantum emitters and metallic nanowires and metal-insulator-metal waveguide structures.
7. • Question: The authors claim that they are the first to show the integration of SPS with photonic circuits. How are the results different from the following papers:
- (a) Deterministic Integration of Single Photon Sources in Silicon-Based Photonic Circuits, Iman Esmaeil Zadeh et al., Nano Lett. 16, 2289-2294, 2016.
 - (b) Strain-Tunable Quantum Integrated Photonics, Ali W. Elshaari et al., Nano Lett. 18, 7969-7976, 2018.
 - (c) Near-Unity Indistinguishability Single Photon Source for Large-Scale Integrated Quantum Optics, Łukasz Dusanowski et al., Phys. Rev. Lett. 122, 173602, 2019.

Furthermore, the authors should also include at the place“It has been shown that nanoscale strain engineering can be used to scale up the creation of such 2D-SPEs [15, 16, 17]”....., some of the relevant recent papers on strain-induced single-photon generation from WSe₂ monolayers and the deterministic coupling of single photons from WSe₂ with plasmonic cavities as follows.

- Kern, J., et al., Nanoscale Positioning of Single- Photon Emitters in Atomically Thin WSe₂, Adv. Mater. 2016, 28, 7101-7105.
 - Laxmi Narayan Tripathi, et al., ACS Photonics 2018 5 (5), 1919-1926. DOI: 10.1021/ac-photonics.7b01053.
 - Oliver Iff, et al., “Deterministic coupling of quantum emitters in WSe₂ monolayers to plasmonic nanocavities,” Opt. Express 26, 25944-25951 (2018).
- Answer: Paper (a) describes the integration of III-V pre-selected quantum dots embedded in nanowires with SiN waveguides. The process described in the paper requires nanowire transfer and etching of a PIC circuit around the nanowire. So this adds considerable challenges in terms of processing and alignment accuracy and moreover requires separate processing of the quantum dot host. We did not claim to be the first to show integration of SPS with photonic circuits in our manuscript, but highlighted the advantages of 2D-emitters over other platforms and moreover explicitly mentioned the issues with III-V based emitters in the Introduction section of the original manuscript (“*This is a major issue for diamond and III-V based quantum technologies, where a separate photonic structure is typically made in the host material to allow efficient single photon transfer between the host and underlying PIC. This adds serious challenges because separate PICs have to be fabricated in the host material and moreover may require precise pick-and-place techniques to integrate both PICs together.*”). Paper (b) describes a similar platform, but now a piezo-electric crystal substrate is added to allow tuning of the emitter wavelength (so even more fabrication challenges are present for this platform). The best tuning efficiency obtained in this paper was 1.33 pm/V. As a rough comparison, C. Chakraborty et al. [4] showed a tuning efficiency of WSe₂ quantum emitters on the order of 10⁴ pm/V. Moreover, this tuning was purely based on electrical means and did not require piezo-electric tuning. So apart from less stringent fabrication, the tuning efficiency can be 10⁴ better as compared to the efficiency shown in paper (b). Paper (c) describes a In(Ga)As/GaAs system where the waveguide is etched in the III-V stack, so this is without SiN. In conclusion, the mentioned papers all describe a substantially different platform to realize single photon emission (all based on III-V systems as opposed to a 2D-material system in our paper). To the best of

our knowledge, our result is still the first to report single photon emission from TMDC-based quantum emitters into the guided mode of CMOS-compatible photonic waveguides. We also added the suggested references to the revised manuscript.

2 Reviewer 2

1. Remarks: In the paper, the authors reported coupling of single quantum emitters in WSe2 to SiN waveguide. They also analyzed performance of a cavity integrated emitter system. I do not find any novelty in the research. WSe2 emitters are very well-studied and integrating with SiN WG is extremely simple. Integration with cavities (experimentally) would have been a significant milestone (due to spectral matching needed), but only theoretical analysis is provided, which is similar to any other emitters. In fact, the authors have published research on similar analysis in the past. Hence, I do not recommend the paper for publication.
2. Answer: We thank the Reviewer for his opinion, but would like to highlight some important facts. While some of the seminal works on nanoscale strain engineering of 2D materials ([5, 6]) clearly highlight the potential of their results for integrated quantum photonics, integration on a CMOS-compatible SiN platform has not been shown so far, despite the fact these works are already 2 years old. We show, to the best of our knowledge, the first experimental demonstration thereof (including a conclusive second order correlation measurement on the integrated emitter, confirming the non-classical nature of the emitted light). Integration with cavities is indeed non-trivial. But apart from the spectral matching (which in principle could be pursued through the electrical tuning as described in the work of one of the co-authors of this paper [4]), it is equally important to study the feasibility of making a resonator that can provide the necessary enhancement for a realistic integrated 2D emitter (e.g. emitter decay rates used in the master equation). As such the presented analysis is definitely relevant for the design of hybrid SiN/TMDC systems. We refined our simulations to even better estimate the decay rates from experimental results, and more clearly discussed the potential large range of quantum efficiencies (as suggested by Reviewer 3). Finally, the other 2 Reviewers found our work interesting and timely for publication given the huge interest in 2D-based emitters.

3 Reviewer 3

In this article, the authors discuss the coupling of single photon emitters originating from a monolayer of WSe2 to a SiN waveguide, a primary component in photonic integrated circuits. Despite a potentially low quantum yield from the single photon emitters, the authors calculate the potential for a high extraction efficiency while indistinguishability remains limited by the SPE quality. This manuscript highlights the ability for monolayer TMDs to be integrated into well-developed PIC processes with a simple transfer step and gives direction for further improvement. Recently there has been a lot of interests in single photon emitters in TMDs and this article is timely for that.

1.
 - Question: The authors show the spectra for spots which specifically contain quantum emitters, but give no spectra off the spots to compare with, could this be included in the supplementary?
 - Answer: We have added a reference spectrum in the Supplementary Information and added a reference to it in the main text.
2.
 - Question: For spot S2, the authors claim that this is near the monolayer edge, however upon examining the optical image closely, one can clearly see that there is a fold in the

WSe2 monolayer and it looks more like a bilayer (perhaps with a twist angle) and an area of high strain which likely is the main contribution to this peak. The authors later comment on this on page 4 without making an overall reference or at small explanation as to why strain contributes, while misleading the reader earlier in the text. Could the authors at least expand on this and give good references, considering this is likely the main mechanism of the quantum emitters that they are coupling to?

- Answer: We have changed the manuscript to incorporate the discussion of strain earlier in the text and added the appropriate references to earlier studies on strain-induced emitters.
- 3.
- Question: Additionally, with the white dots covering the exact area of the spots examined, it is hard to tell from Figure 2b what the integrated intensity at the exact spots looks like compared to nearby or the background. Finally, from Figure 2, it seems to be that the highest intensity in the waveguide PL scan (c) comes from areas not over the spots (sans S3). Could the authors comment on this?
 - Answer: We have changed the white dots by white asterisks for increased visibility of the integrated PL background. The areas near the the edges of the waveguide are more likely to have higher strain, which is beneficial for the creation of strain-induced emitters. On the other hand, the overlap between a dipole emitter and the waveguide mode will be highest at the center of the waveguide (where the evanescent field is maximum). So therefore one would in general expect that the 2D material couples more efficiently near the center of the waveguide, resulting in an overall higher integrated intensity. We have added this comment in the main manuscript as well.
- 4.
- Question: In the main text γ_p is completely left undefined , and this happens to a few other variables as well. Although many of these variables are defined in the supplementary, there should be at least some clarification in the main text, given that there are a particularly large number of variables in this text and it is incredibly easy for the reader to become confused, especially those from a broader audience as expected from a journal like Nature Communications.
 - Answer: We thank the Reviewer for pointing this out and have added clarifications on the variables where needed.
- 5.
- Question: The authors use spot 5 to examine the quality of their emitter/waveguide, why specifically this spot?
 - Answer: For this particular location the spectrum contains a fairly isolated and prominent peak around 1.64 eV (756.5 nm). Since the overall intensity of the emitters is still relatively low, we want to avoid losses as much as possible when performing a second-order correlation measurement. Instead of filtering out this line with a monochromator, we used a 750 nm longpass filter to isolate the line from the rest of the spectrum and simultaneously minimize transmission losses to the single photon detectors. While this adds some background to the signal, it does allow us to maintain the highest possible count rate and correct for the small background afterwards (this process is described both in the main text and the supplementary material).
- 6.
- Question: The authors used a reported value for the quantum yield in monolayer WSe2 to further calculate a decay rate of the emitter to all modes, this value is quoted as 3%. However, this is a vastly incorrect value to use as it takes into account the entire PL spectrum as discussed in reference 39, not the individual emitter. Secondly, in reference 39 the material is purchased from 2 separate companies which utilize CVT to produce the single crystals. These crystals can vary dramatically in quality in a singular batch (there

are at least 4 main growth parameters which govern this process) making any comparison of quality to this work impossible as QY's could vary by orders of magnitude between samples extracted from different crystals in the same batch. This is not at all hinted in the calculation or the Supplementary.

- Answer: We added a modified reference (on the QY of localized emitters in 2D TMDCs) and commented on the fact that the QY can vary significantly. We moreover refined some of the calculations and adapted the numerical values in the main text as such.

References

- [1] Chen, J., Smolyakov, G.A., Brueck, S.R.J., Malloy, K.J., Surface plasmon modes of finite, planar, metal-insulator-metal plasmonic waveguides, *Opt. Express*, **16(19)**, 14902–14909 (2008).
- [2] Walls, D.F., Milburn, G.J., Quantum Optics (2nd Edition). (Springer-Verlag Berlin Heidelberg, 2008).
- [3] Luo, Y., et al., *Nat. Nanotech.*, **13**, 1137–1142 (2018).
- [4] Chakraborty, C., Goodfellow, K.M., Dhara, S., Yoshimura, A., Meunier, V., Vamivakas, A.N., Quantum-Confined Stark Effect of Individual Defects in a van der Waals Heterostructure, *Nano. Lett.*, **17**, 2253–2258 (2017).
- [5] Branny, A., Kumar, S., Proux, R., Gerardot, B.D., Deterministic strain-induced arrays of quantum emitters in a two-dimensional semiconductor, *Nat. Comm.*, **8**, 15053 (2017).
- [6] Palacios-Berraquero, C., et al., Large-scale quantum-emitter arrays in atomically thin semiconductors, *Nat. Comm.*, **8**, 15093 (2017).

REVIEWERS' COMMENTS:

Reviewer #1 (Remarks to the Author):

Authors have answered all my questions and incorporated the comments/suggestions in the manuscript. The manuscript is improved significantly, therefore, I recommend for its publication.

Reviewer #3 (Remarks to the Author):

The authors have sufficiently addressed all of my concerns.